# Effect of Curcumin Pretreatment on the Susceptibility of *Cryptococcus neoformans* to Photodynamic Therapy Mediated by Aluminum Phthalocyanine in Nanoemulsion

**DOI:** 10.3390/ph18020240

**Published:** 2025-02-11

**Authors:** Fabiana Chagas Costa, Lourival Carvalho Nunes, Kunal Ranjan, Ariane Pandolfo Silveira, Ingrid Gracielle Martins da Silva, André de Lima e Silva Mariano, Paulo Eduardo Narcizo de Souza, Sônia Nair Báo, Marcio Jose Poças-Fonseca, Luis Alexandre Muehlmann

**Affiliations:** 1Laboratory of Nanoscience and Immunology, Faculty of Health Sciences and Technologies, University of Brasilia, Campus Ceilandia, Brasília 72220-900, Brazil; fchagas16@gmail.com; 2Department of Genetics and Morphology, Institute of Biological Sciences (IB), University of Brasilia (UnB), Brasilia 70910-900, Brazil; lourivalcarvalho61@gmail.com (L.C.N.); kukkukr.ranjan@gmail.com (K.R.); 3Amity Institute of Biotechnology, Amity University Jharkhand, Ranchi 834001, India; 4Laboratory of Microscopy and Microanalysis—LMM, Department of Cell Biology, Institute of Biological Sciences (IB), University of Brasilia (UnB), Brasilia 70910-900, Brazil; pandolfo.ariane@gmail.com (A.P.S.); gracilias@gmail.com (I.G.M.d.S.); snbao@unb.br (S.N.B.); 5Laboratory for Softwares and Physics Instrumentation Development, Institute of Physics, University of Brasilia (UnB), Brasilia 70910-900, Brazil; demariano95@gmail.com (A.d.L.e.S.M.); psouza@unb.br (P.E.N.d.S.)

**Keywords:** cryptococcosis, epigenetics, histone deacetylase inhibitors, nanotechnology, pathogenic fungi

## Abstract

**Background/Objectives:** Curcumin has antimicrobial activity, and its mechanism of action involves changing histone acetylation. Our group has shown that histone deacetylases (HDACs) inhibitors increase the sensibility of *Cryptococcus neoformans* to certain antifungal treatments. Therefore, the aim of this work was to investigate whether curcumin pretreatment increases the effect of photodynamic therapy (PDT) mediated by aluminum phthalocyanine in nanoemulsion (AlPc-NE) against *C. neoformans*. **Methods:** The minimum inhibitory concentrations (MIC) of AlPc-NE and curcumin, along with the 72-h growth curve of cells exposed to the combined treatments, were evaluated in the *C. neoformans* reference strain H99. Additionally, further analysis was performed using HDAC gene deletion mutant strains, *hda1*Δ and *hos2*Δ. **Results:** Curcumin reduces the effect of PDT on *C. neoformans* reference strain H99, likely due to its antioxidant properties. In the *hda1Δ* strain, 50% MIC of curcumin reduced the effect of PDT, but this effect was not observed in response to 75% MIC of curcumin. Conversely, in the *hos2*Δ strain, pretreatment with curcumin at 75% MIC enhanced the efficacy of PDT in combination with 50% MIC of AlPc-NE. **Conclusions:** These results indicate that curcumin inhibits *C. neoformans*. Moreover, at lower concentrations, curcumin protects cells against oxidant damage, while at higher concentrations, it may trigger epigenetic mechanisms that compromise cell viability. In conclusion, both curcumin and PDT are active against *C. neoformans*, with HDACs affecting their efficacy, and the effectiveness of the combined treatment depends on the concentration of both curcumin and AlPc-NE.

## 1. Introduction

In 2022, the World Health Organization (WHO) published the first list of priority fungal pathogens to guide research, surveillance, and public policy efforts. Leading the list is *Cryptococcus neoformans*, the most common cause of cryptococcosis, a globally prevalent systemic mycosis that primarily affects immunocompromised individuals [1]. One of the main limitations in the treatment of cryptococcosis is the sole reliance on chemotherapy, which is based on three of the four families of available systemic antifungal drugs: azoles, polyenes, and pyrimidines. Amphotericin B remains the first-line treatment, despite its toxicity, often in combination with flucytosine or fluconazole. However, a 12% incidence of fluconazole-tolerant isolates has already been reported, as reviewed elsewhere [2].

In the search for new approaches against *C. neoformans*, our group has shown that the efficacy of photodynamic therapy (PDT) is significantly increased by pretreating the cells with either sodium butyrate or tricostatin A, both inhibitors of histone deacetylases (HDACs) [3]. PDT is based on the generation of reactive species following the photoactivation of a photosensitizer within the target cell and has been used for several biological applications [4,5,6]. Using HDAC gene deletion mutant strains, *hda1*∆ and *hos2*∆, our group has further evidenced that HDACs protect *C. neoformans* against PDT and other antifungal treatments [7]. HDACs catalyze the hydrolysis of acetyl groups from post-translationally modified lysine residues in histone proteins, increasing the net positive charge of these proteins and therefore enhancing their affinity to DNA. This process leads to chromatin condensation, which typically results in gene repression, as the tighter chromatin structure makes it more difficult for the transcriptional machinery to access DNA [8]. Therefore, HDACs play a critical role in chromatin remodeling and gene expression regulation, influencing fungal pathogenesis, stress responses, and drug resistance mechanisms [8]. Studies have shown that HDAC inhibitors can modulate fungal virulence traits, including biofilm formation and resistance to oxidative stress, suggesting their potential as antifungal adjuvants [7,9]. Thus, these epigenetic modulators represent potential therapeutic targets in *C. neoformans*, and the discovery of effective HDAC inhibitors offers promising prospects for the future treatment of cryptococcosis.

In this context, curcumin has been pointed out as a potent inhibitor and suppressor of HDAC expression in cancer cells, likely as a result of its ability to modulate epigenetic regulation by altering histone acetylation patterns [10,11,12]. Curcumin has been shown to increase histone acetylation levels by inhibiting HDAC enzymatic activity, leading to chromatin relaxation and transcriptional activation of genes involved in stress responses, apoptosis, and cell cycle regulation [13]. Additionally, curcumin may influence HDAC activity indirectly by modulating signaling pathways such as those involving mitogen-activated protein kinase (MAPK) and phosphoinositide 3-kinase/protein kinase B (PI3K/PKB), which are known to regulate epigenetic modifiers [14]. These mechanisms suggest that curcumin could similarly affect fungal HDACs, potentially altering the gene expression networks that control virulence, stress adaptation, and antifungal resistance in *C. neoformans*. Although the literature indicates that curcumin is active against *Cryptococcus* spp. [15,16], there is a scarcity of studies focusing on its effects on the epigenetic machinery of fungi. Moreover, the use of curcumin to potentiate PDT against *C. neoformans* has not yet been investigated. Our hypothesis is that curcumin may affect HDACs activity in *C. neoformans*, and thus may be able to potentiate the effect of PDT against this pathogen. Therefore, this study aimed to evaluate whether curcumin enhances the effect of AlPc-NE-mediated PDT against *C. neoformans* and to explore the involvement of HDACs using the *hda1*∆ and *hos2*∆ mutant strains.

## 2. Results and Discussion

### 2.1. Curcumin MIC for C. neoformans Strains

MICs of curcumin for *C*. *neoformans* strains are shown in Table 1. This result shows that curcumin has antifungal activity and corroborates previous studies that evidenced the antifungal activity of curcuminoids against *Cryptococcus* spp. [15,16]. However, no statistically significant difference for this variable was found between the strains.

### 2.2. MIC of AlPc-NE in PDT Against C. neoformans Strains

The effect of PDT mediated by AlPc-NE was evaluated against *C. neoformans* strains. HDAC mutants were more susceptible (MIC of 3 nM) to PDT than the reference strain (MIC of 6 nM), as shown in Table 2. A previous study by our group has demonstrated these *C. neoformans* strains to be susceptible to PDT, with AlPc-NE MIC values ranging from 6.25 to 12.5 nM [7]. The fact that MIC values for *C. neoformans* are consistently in the nanomolar range shows that this pathogen is particularly sensitive to AlPc-NE-mediated PDT. This, together with the fact that PDT’s effects can be limited to the irradiated site, with minimal side-effects in non-irradiated, non-target tissues [17], points to the possibility that this PDT protocol could be used to treat certain manifestations of cryptococcosis, such as those in the lungs, skin, and eyes.

The HDAC mutant strains were more susceptible to PDT treatment than the reference strain, indicating that HDACs are involved in cellular defenses against oxidative damage caused by this therapy, as previously suggested [3,7,18]. Therefore, the use of HDAC inhibitors as pretreatments is a possible approach to increasing the susceptibility of *C. neoformans* to PDT [3].

Regarding the higher susceptibility of HDAC gene deletion strains, our results corroborate the findings described by Ranjan and collaborators [7,18]. In 2021, Ranjan et al. [7] investigated the role of *C. neoformans* HDAC genes in its response to antifungal drugs, sodium butyrate or trichostatin A—classical epigenetic modulators—and PDT. They found that inhibition of HDAC genes increases the susceptibility of the pathogen to PDT and drug treatments, highlighting their potential as therapeutic targets for enhancing efficacy against *Cryptococcus* spp. In 2024, Ranjan et al. [18] reported that the effects of PDT against *Cryptococcus* and *Candida* species are enhanced by *Streptomyces* spp. extracts, suggesting a potential synergistic approach for treating fungal infections in vitro. This effect was more pronounced in *hda1*Δ and *hos2*Δ, the HDAC gene deletion mutant strains of *C. neoformans*. In the present work, our goal was to verify whether pretreatment with curcumin, a potential HDAC inhibitor in *C. neoformans*, could increase the susceptibility of *C. neoformans* to PDT. To our knowledge, this is the first study to investigate this approach.

### 2.3. Combined Effect of Curcumin and PDT

The prior exposure of yeast to MIC_50_ and MIC_75_ of curcumin, followed by the application of PDT mediated by MIC_50_ and MIC_75_ of AlPc-NE, led to a significant decrease in cell proliferation (Figure 1). Inhibition of cell growth was observed in all tested combinations, but not in the individual use of curcumin at values below the MIC.

The pretreatment with curcumin followed by PDT had a protective effect, since the yeasts started to grow at around 42 h (Figure 1). This growth is delayed in comparison to that of cells treated with curcumin only, but a stronger effect was observed in response to PDT alone, which abolished cell growth for the tested time. This suggests that curcumin can act as an antioxidant agent at the concentrations tested in the reference H99 strain of *C. neoformans*, possibly by neutralizing oxidant species generated by the photoactivated phthalocyanine. The antioxidant activity of curcuminoids is widely reported in the literature [19]; thus, this protective effect was expected, considering the pro-oxidant nature of PDT.

The possible photosensitizing effect mediated by curcumin in these experiments can be disregarded, as this compound does not absorb light at λ 660 nm, the wavelength used in the PDT protocol in this work. Curcumin, as a photosensitizer, requires violet/blue light, with wavelengths normally within the range from 408 to 434 nm, as reviewed elsewhere [20]. The absence of curcumin-based photodynamic activity can be further confirmed by our results, as H99 cells exposed to curcumin then irradiated with red light, i.e., irradiated 50% CURC and 75% CURC, did not show any decrease in their growth in comparison to the control (Figure 1).

The results presented in Figure 2 and Figure 3 for the mutant *C. neoformans* strains *hda1*Δ and *hos2*Δ, respectively, corroborate our data on the MIC of curcumin, suggesting that the mechanism of action of this compound involves alterations in HDAC activity. The protective effect of curcumin against PDT, observed in the reference strain H99, was not seen in certain conditions in the mutants.

At MIC_50_, curcumin slightly protected *hda1*Δ cells against PDT, as observed for the H99 strain, an effect that is probably due to its antioxidant activity. However, at MIC_75_ curcumin did not protect *hda1*Δ cells against PDT. This might seem paradoxical, as at higher concentrations, one can expect curcumin to have a more pronounced antioxidant effect and thus exert a proportionally more intense protective effect against the oxidant stress generated by PDT. However, this finding can be explained by other mechanisms by which curcumin affects cell metabolism, which are not related to redox events. Indeed, as previously shown in cancer cells [11] and in *Cryptococcus* spp. [15,16], curcumin exerts inhibitory effects at specific concentrations.

Different results were observed in *hos2*Δ. Curcumin at MIC_50_ did not alter the viability of cells subjected to PDT mediated by either the MIC_50_ or the MIC_75_ of AlPc-NE. The same was observed for curcumin at MIC_75_, which did not protect cells against PDT mediated by AlPc-NE at MIC_75_. However, curcumin at MIC_75_ increased the effectiveness of PDT mediated by AlPc-NE at MIC_50_.

Taken together, these results suggest that *hda1* and *hos2* genes are involved in the cellular pathways that modulate the effects of the pretreatment with curcumin, including its antioxidant and potentially non-redox-related mechanisms, in the tested PDT protocols. The absence of the *hda1* gene increased the protective effect of curcumin in *C. neoformans*, suggesting that the enzyme histone deacetylase 1 increases the sensitivity of *C. neoformans* to the inhibitory effects of curcumin. This conclusion is corroborated by the higher MIC exhibited by *hda1*Δ strain (Table 1). The enzyme histone deacetylase 2, coded by the *hos2* gene, seems to play a less significant role in the response of *C. neoformans* to low concentrations of curcumin. The deletion of the *hos2* gene even increased the susceptibility of *C. neoformans* to curcumin at MIC_75_, as observed by the increased inhibition when combined with PDT mediated by AlPc-NE at MIC_50_. Indeed, a lower curcumin MIC was observed for *hos2*Δ (Table 1).

## 3. Material and Methods

### 3.1. Reagents and Equipment

Curcumin from *Curcuma longa* (65%) was purchased from Sigma Aldrich (St. Louis, MO, USA). Dextrose, aluminum phthalocyanine chloride, Cremophor ELP^®^, castor oil, NaCl and Roswell Park Memorial Institute (RPMI) medium were purchased from Sigma Aldrich (St. Louis, MO, USA). Dimethyl sulfoxide (DMSO) was purchased from Dinamica (Indaiatuba, São Paulo, Brazil). K_2_HPO_4_ and KH_2_PO_4_ were purchased from Quimex (Uberaba, Minas Gerais, Brazil). Yeast extract and peptone were purchased from Kasvi (Roseto degli Abruzzi, Terano, Italy). The spectrophotometer/incubator (Epoch 2 EON Microplate reader) was purchased from Biotek Inc, Kaysville, UT, USA. The LED (light-emitting device) was developed by Prof. Paulo Eduardo Narciso de Souza and collaborators (Laboratory of Software and Instrumentation of the Institute of Physics of the University of Brasilia, Brasilia, Brazil).

### 3.2. Strains and Growth Conditions

The wild-type *C. neoformans* H99 strain was generously provided by Dr. Joseph Heitman Laboratory (Duke University Medical Center, Durham, NC, USA). The *hda1*∆ and *hos2*∆ strains were generated and phenotypically characterized by our group [9]. The cells were stored as 35% glycerol stocks and kept at −80 °C. For the experiments, the strains were thawed and cultured on YPD agar plates at 30 °C for 3 days.

### 3.3. Assessment of Curcumin MIC

Stock solutions of curcumin in 100% DMSO were kept frozen at −20 °C until use. For MIC assessment, the NCCLS M27-A3 protocol was employed [21]. According to this method, curcumin stock was diluted in 2-fold series to obtain test concentrations (1000.0 µg/mL, 500.0 µg/mL, 250.0 µg/mL, 125.0 µg/mL, 62.5 µg/mL 31.2 µg/mL 15.6 µg/mL 7.8 µg/mL, 3.9 µg/mL, 1.9 µg/mL and 1.0 µg/mL). After this experiment, we tested further concentrations around the initial MIC (ranging from 1.0 to 30.0 µg/mL). Positive controls, which lacked extracts, and negative controls, which lacked yeast cells, were utilized.

Yeast cells were grown in YPD broth at 30 °C with agitation for 20 h. After washing with PBS, their concentration was adjusted to 4 × 10^4^ cells/mL in RPMI.

One hundred microliters (μL) of the cell suspension was added to 96-well plates then treated with curcumin. Plates were visually assessed after 72 h of incubation at 37 °C under orbital shaking at 150 rpm. Three independent experiments were performed for each experimental condition. The MIC was defined as the lowest concentration of curcumin that completely inhibited fungal growth in a 96-well polystyrene plate.

### 3.4. Assessment of AlPc-NE MIC in PDT

Different concentrations of the photosensitizer AlPc-NE were initially tested (1.6, 3.1, 6.2, 12.5, 25.0, 50.0, 100.0, 200.0, 300.0 or 400.0 nM AlPc-equivalent), followed by refinement with concentrations ranging from 1.0 to 20.0 nM. After treatment with AlPc-NE, the cells were incubated in the dark at 28 °C under agitation on an inclined platform at 50 rpm for 30 min, as previously described [3,7,18]. Then, the cells were centrifuged for 5 min at 12,000 rpm, washed in PBS to remove non-internalized AlPc-NE, and resuspended in RPMI medium with the appropriate concentrations. The cells were then plated in two 96-well polystyrene microplates. One of the plates was irradiated with LED at a wavelength of 660 nm and a power of 59.71 mW/cm^2^ for 10 min with a fluence of 35.83 J/cm^2^.

The microplates were incubated in the Epoch^®^ 2 EON microplate reader with orbital shaking for 72 h at 37 °C. Absorbance at 600 nm was measured to assess cell proliferation, according to Ranjan et al. (2024) [18]. The MIC was defined as the lowest concentration of AlPc-NE that completely inhibited fungal growth in this PDT protocol.

### 3.5. Combined Effect of PDT and Curcumin

The impact of PDT associated with pre-exposure to curcumin on the growth of *C*. *neoformans* was evaluated as previously established [18]. Briefly, *C. neoformans* cells were grown in 2 mL of liquid YPD medium with 50% and 75% of the previously determined curcumin MIC at 28 °C for 24 h. After incubation, cells were washed with PBS and the concentration was adjusted to 2 × 10^5^ cells/mL. The cells were then treated with 50% and 75% of the AlPc-NE MIC and incubated at 28 °C, 50 rpm for 30 min. Cells were washed and resuspended in RPMI medium and added in two 96-well polystyrene plates, then one was irradiated with LED under the aforementioned conditions.

Both plates were then incubated at 37 °C under orbital shaking in a spectrophotometer/incubator as mentioned above.

### 3.6. Statistical Analyses

Graphs and two-factor analysis of variance (ANOVA) were performed using GraphPad Prism 9.0 software (GraphPad Software, San Diego, CA, USA). The significance level adopted was *p* < 0.05. Cell growth curves were obtained and compared with the respective controls. The results were expressed as the mean ± standard error of the mean, and the means of technical triplicates from three independent experiments were statistically evaluated. Line graphs were generated to illustrate the microorganism growth curves, while histograms were used to represent the area under the curve.

## 4. Conclusions

This study examined the effects of curcumin and PDT, used individually or in combination, on the growth of *C. neoformans* strains, including both wild-type and HDAC gene deletion mutants. Both curcumin and PDT inhibit *C. neoformans* and have significant potential as therapeutic agents for controlling cryptococcosis. Each treatment used individually was effective in inhibiting *C. neoformans* growth, and this work provides evidence that the HDAC genes *hos2* and *hda1* are involved in the response of this pathogen to these approaches. The efficacy of the combined treatment, however, depends on the concentration of both the photosensitizer AlPc-NE and curcumin.

These findings support the continued investigation and development of this approach as a potential option for treating fungal infections caused by *C. neoformans*, with the potential to improve the efficacy of existing treatments and address the growing antifungal resistance.

## Figures and Tables

**Figure 1 pharmaceuticals-18-00240-f001:**
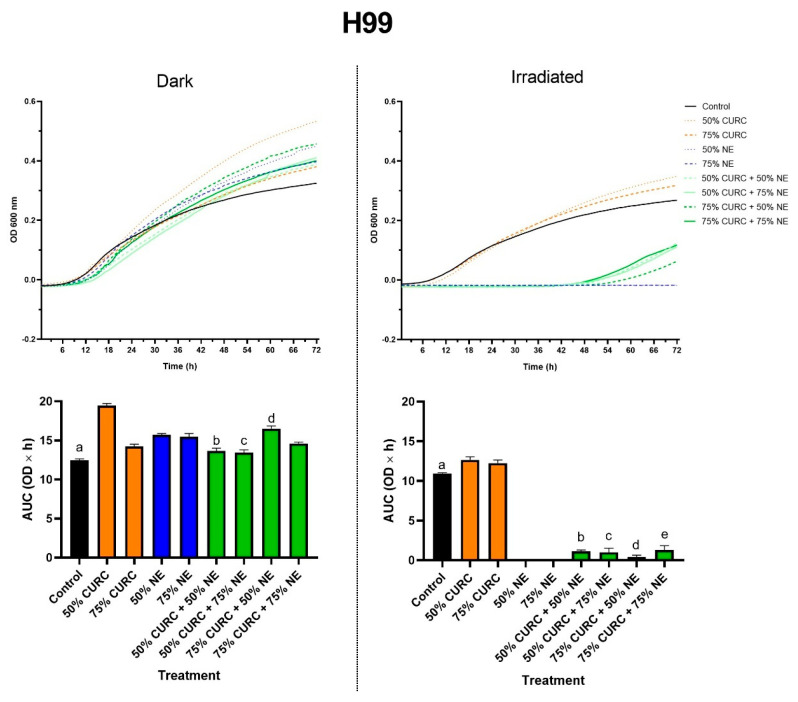
Growth of *Cryptococcus neoformans* H99 strain after exposure to curcumin (CURC) and/or nanoemulsion containing aluminum phthalocyanine (NE), followed by incubation in the dark or irradiation with red light (660 nm, 35.83 J/cm^2^). The fungi were exposed to 50 or 75% of the minimum inhibitory concentration of each compound, alone or in combination. The upper row shows the growth curves as optical density at 600 nm (OD 600 nm) over time in hours. The lower row shows the graphs with the areas under these growth curves (AUC, OD × h) for each group. Statistical significance: dark—a *p* < 0.05 vs. all groups; b *p* < 0.001 vs. 50% CURC, and vs. 50% NE; c *p* < 0.01 vs. 75% NE; d *p* < 0.05 vs. 75% CURC. Irradiated—a *p* < 0.01 vs. all groups; b *p* < 0.001 vs. 50% CURC; c *p* < 0.001 vs. 50% CURC and 50% NE; d *p* < 0.001 vs. 75% CURC and *p <* 0.05 vs. 50% NE; e *p* < 0.001 vs. 75% CURC and 75% NE.

**Figure 2 pharmaceuticals-18-00240-f002:**
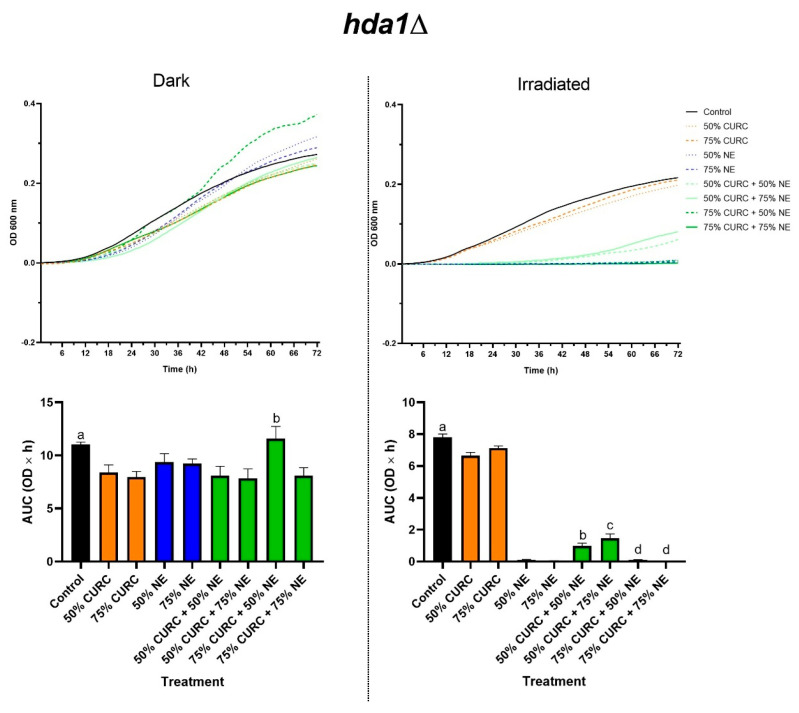
Growth of *C. neoformans*, *hda1Δ* mutant, after exposure to curcumin (CURC) and/or nanoemulsion containing aluminum phthalocyanine (NE), followed by incubation in the dark or irradiation with red light (660 nm, 35.83 J/cm^2^). The fungi were exposed to 50 or 75% of the minimum inhibitory concentration of each compound, alone or in combination. The upper row shows the growth curves as optical density at 600 nm (OD 600 nm) over time in hours. The lower row shows the graphs with the areas under these growth curves (AUC, OD × h) for each group. Statistical significance: Dark—a *p* < 0.01 vs. 50% CURC, 75% CURC, 50% CURC + 75% NE, and 75% CURC + 75% NE; b *p* < 0.01 vs. 75% CURC, and 50% NE. Irradiated—a *p* < 0.01 vs. all groups, except 75% CURC; b *p* < 0.001 vs. 50% CURC, and 50% NE; c *p* < 0.001 vs. 50% CURC, and 75% NE; d *p* < 0.001 vs. 75% CURC.

**Figure 3 pharmaceuticals-18-00240-f003:**
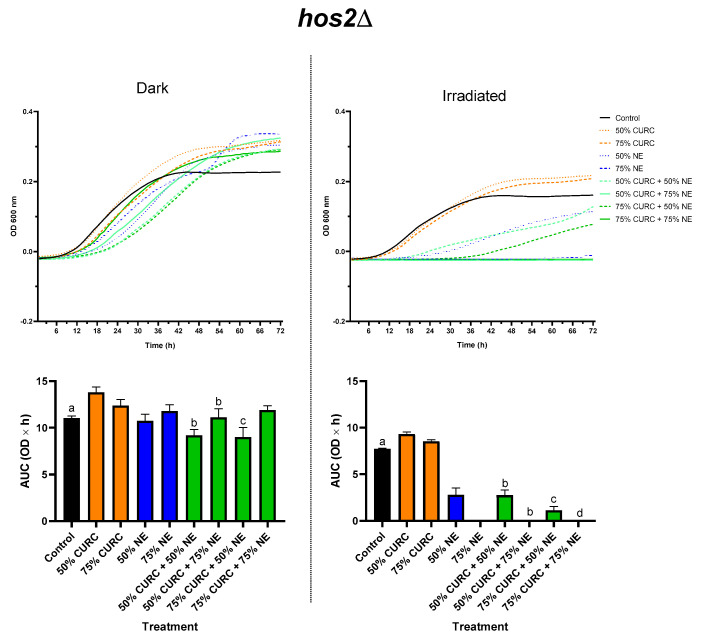
Growth of *C. neoformans*, *hos2Δ* mutant, after exposure to curcumin (CURC) and/or nanoemulsion containing aluminum phthalocyanine (NE), followed by incubation in the dark or irradiation with red light (660 nm, 35.83 J/cm²). The fungi were exposed to 50 or 75% of the minimum inhibitory concentration of each compound alone or in combination. The upper row shows the growth curves as optical density at 600 nm (OD 600 nm) over time in hours. The lower row contains graphs with the areas under these growth curves (AUC, OD × h) for each group. Statistical significance: Dark—a *p* < 0.05 vs. 50% CURC and 75% CURC + 50% NE; b *p* < 0.01 vs. 50% CURC; c *p* < 0.01 vs. 75% CURC, and 75% NE. Irradiated—a *p* < 0.0001 vs. all groups, except 75% CURC; b *p* < 0.001 vs. 50% CURC; c *p* < 0.001 vs. 75% CURC; d *p* < 0.001 vs. 75% CURC.

**Table 1 pharmaceuticals-18-00240-t001:** Minimum inhibitory concentrations (MICs) of curcumin against different strains of *C. neoformans* (reference strain H99, and *hda1*Δ and *hos2*Δ mutants). Values are presented as the mean ± standard error of the mean (SEM).

Strain	MIC (μg/mL)
H99	21.7 ± 0.3
*hda1*Δ	24.7 ± 3.2
*hos2*Δ	19.7 ± 0.3

**Table 2 pharmaceuticals-18-00240-t002:** Minimum inhibitory concentrations (MICs) of aluminum phthalocyanine chloride nanoemulsion against different strains of *C. neoformans* (reference strain H99, and *hda1*Δ and *hos2*Δ mutants). Values are presented as the mean ± standard error of the mean (SEM).

Strain	MIC (nM)
H99	6.3 ± 0.3 *
*hda1*Δ	3.0 ± 0.0
*hos2*Δ	3.3 ± 0.3

* *p* < 0.001 vs. *hda1*Δ and vs. *hos2*Δ.

## Data Availability

The original contributions presented in this study are included in the article. Further inquiries can be directed to the corresponding authors.

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
