# Peer review of "Effect of Curcumin Pretreatment on the Susceptibility of Cryptococcus neoformans to Photodynamic Therapy Mediated by Aluminum Phthalocyanine in Nanoemulsion"

_pharmaceuticals, 2025, doi:10.3390/ph18020240_

Round 1

Reviewer 1 Report

Comments and Suggestions for Authors

Comments to authors

The research paper entitled “Combined effect of curcumin and photodynamic therapy medi- ated by aluminum-phthalocyanine in nanoemulsion against Cryptococcus neoformans strains is very interesting approach for controlling fungi. The manuscript presentation is very good. This research work is valuable and merit publishing in “Pharmaceuticals” after some minor corrections.

1.      Revise the title in good manner.

2.      The abstract should be revised.

3.      Conclusive remarks are missing in abstract. Please add that sentence.

4.      Revise the keywords and arrange in alphabetical order. The keywords highlight the conducted research. These should be more meaningful the readers’s attraction.

5.      The introduction section is not sufficient. Add more related information from literature.  

6.      Improve the resolution of all the figures in the manuscript.

7.      The references in manuscript text should be cited according to journal format.

8.      hda1Δ……This is bacteria name or something else please clarify?

9.      Line 217: One hundred μL….Add microliter before μL

10.  Check the references. These should be in same style.

11.  Conclusion: This study investigated the action of individual and combined treatments in control- 256 ling the growth of C. neoformans strains, both wild-type and mutants with deleted HDAC- 257 encoding genes….Revise this sentence

12.  Tools….Add agents instead of tools

13.  How this research work is commercialized for the benefit of general public?

14.  The manuscript should be revised carefully according to journal’s format. There are a lot of minor but basic grammatical mistakes in the manuscript. These mistakes should be removed.

15.  The Plagiarism of the manuscript should be checked after revision.

Author Response

Dear reviewer, 

We thank you very much for your attention towards our manuscript. Your comments surely helped to improve the presentation of our report. All changes are now highlighted in the revised version of the manuscript.

Plase find below each of the raised points addressed individually:

  1. Revise the title in good manner.A.: the title was modified accordingly.

2.      The abstract should be revised.

A.: the abstract was altered to make it clearer.

3.      Conclusive remarks are missing in abstract. Please add that sentence.

A.: a conclusion was added to the abstract.

4.      Revise the keywords and arrange in alphabetical order. The keywords highlight the conducted research. These should be more meaningful the readers’s attraction.

A.: we have ordered the keywords alphabetically. We think that the keywords must be different from those in the title, as both the title and keywords are used by search tools to find the article. We kindly ask the reviewer to consider this.

5.      The introduction section is not sufficient. Add more related information from literature. 

A.: we have added more information to the introduction as requested.

6.      Improve the resolution of all the figures in the manuscript.

A.: all the figures are now presented in the text in the highest resolution generated possible using Graphpad Prism software. 

7.      The references in manuscript text should be cited according to journal format.

A.: The citation is now correct. 

8.      hda1Δ……This is bacteria name or something else please clarify?

A.: this is the symbol used to describe a mutant produced by deletion of the hda1 gene from C. neoformans wild-type strain. So, hda1: refers to the HDA1 gene, which encodes a histone deacetylase (HDAC). Δ : Signifies a deletion of the gene. These symbols are defined in the introduction and methods sections. 

9.      Line 217: One hundred μL….Add microliter before 

A.: alteration done.

10.  Check the references. These should be in same style.

A.: references have been checked and are in the style demanded by Pharmaceuticals.

11.  Conclusion: This study investigated the action of individual and combined treatments in control- 256 ling the growth of C. neoformans strains, both wild-type and mutants with deleted HDAC- 257 encoding genes….Revise this sentence

A.: this sentence was revised, as highlighted on the text: "This study examined the effects of curcumin and PDT, used individually or combined, on the growth of C. neoformans strains, including both wild-type and HDAC gene deletion mutants."

12.  Tools….Add agents instead of tools

A.: correction done.

13.  How this research work is commercialized for the benefit of general public?

A.: as cited in the conclusion, our work has "the potential to improve the efficacy of existing treatments and address the growing antifungal resistance." This would benefit people suffering form cryptococcosis provided this technology becomes a product or a protocol in the future.

14.  The manuscript should be revised carefully according to journal’s format. There are a lot of minor but basic grammatical mistakes in the manuscript. These mistakes should be removed.

A.: The manuscript has been now checked again for grammar and typpos.  

15.  The Plagiarism of the manuscript should be checked after revision.

A.: The manuscript has been now checked again for plagiarism wiuth Turnitin provided by our institution.  

Reviewer 2 Report

Comments and Suggestions for Authors

Dear respected Authors

Overall, your manuscript is a good presentation with describing results and discussion of a new investigation in curcumin effect and PDT by AIPc-NE against C. neuformans. I give several comments and need clarification about your experiment for better quality in your publication. Please check my comment here

Author Response

Dear reviewer, we, the authors, thank you for your precious contribution towards our work.

Please find below the responses to each of the points you raised:

  • Abstract: the demanded changes were performed and are highlighted in the re-submited manuscript.
  • Introduction: we have read this topic again and, in our humble opinion, our hypothesis and its fundaments are cleary stated. We kindly ask you to take into account the following text in the last paragraph of Introduction which expresses: "Although literatures indicate that curcumin is active against Cryptococcus spp. (11,12), there is a scarcity of studies focusing on its effects on the epigenetic machinery of fungi. Moreover, the use of curcumin to potentiate PDT against C. neoformans has not yet been investigated. Our hypothesis is that curcumin may affect HDACs activity in C. neoformans, thus being able to potentiate the effect of PDT against this pathogen." As stated here, the use of curcumin as a pretreatment for PDT and the role of HDACs in the mechanism of action of curcumin against C. neoformans have not yet been reported. To our knowledge, our work is the first to approach these topics.
  • Tables 1 and 2: we have performed 3 independent experiments for MIC of both PDT and curcumin. We presented only the average, rounded to the nearest whole number. Now the tables show the mean with the standard error of the mean.
  • Regarding the comment: "Table 1 suggest that the hos2 gene reduces, while the hda1 gene increases. How about with H99 ? you should highlight scientifically why the hos2 gene, while hda1 increase?" we have added the following discussion: "This can be a result of changes in chromatin compatation, which could modify the susceptibility of the DNA do damage, or modify gene expression altering the sensibility of the cells to the effects of curcumin." to the manuscript. The discussion of the results was also modified to approach the presented results. Thank you for this comment.
  • Regarding the comment: "This study is almost similar with Ranjan et al with using HDAC mutant strains. Please clarify !" Our group as focused on the prospection of HDACs inhibitors for combinatory antifungal treatments. The previous studies by Ranjan, published in 2021 and in 2024, had different goals. The first one reported that classical HDAC inhibitors, such as butyrate, increase the susceptibility of Cryptococcus spp to antifungals such as PDT and fluconazol. The second one focused on prospecting possible HDAC inhibitors from Streptomyces spp. extracts, which have never been studied for their HDAC inhibition potential. The present manuscript explores the potential of curcumin to act as a compound that alters HDAC activity thereby modifying the susceptibility of C. neoformans to PDT. This curcumin's acitivity has never been studied in this fungi, and therefore our study is original in the literature, even considering previous works our group have published. We humbly ask the steemed reviewer to notice that our group has been prospecting HDAC inhibitors, and curcumin is a potential candidate as indicated by studies with mammal cancer cells.
    We have added this discussion to the topic "MIC of AlPc-NE in PDT against C. neoformans strains" in "Results and discussion" section.
  • Regarding the comment in the legend of Figure 1: we have added the reference to using OD 600 nm in the methods section, "3.4. Assessment of AlPc-NE MIC in PDT".
  • Regarding the comment in Figure 1: we have added a line separating left (Dark) and right (Irradiated) sides of figures. We think that this is the best way to express these results, as separating it in two different figures would make it difficult to see the effect of iradiation itself on PDT protocols.
  • We have added the required references in the methods section. 

We thank you for your attention towards our manuscript. We think that after adressing the points raised by you, the quality of our report improved significantly. 

Reviewer 3 Report

Comments and Suggestions for Authors

The aim of this work was to investigate whether curcumin pretreatment increases the effect of photodynamic therapy mediated by aluminum-phthalocyanine in nanoemulsion (AlPc-NE) against C. neoformans. The article seems to be very interesting. 

However, I think some revisions are needed.

The description of the results is not clear in the abstract and the full term (PDT), photodynamic therapy, should be inserted.

The authors do not explain why curcumin was chosen. The introduction should expand the literature on the subject, describe in more detail the activity of curcumin as an epigenetic modulator and report other examples of natural substances with similar action.

The results presented in table 1 suggest that the hos2 gene reduces, while the hda1 gene increases, the susceptibility of C. neoformans to curcumin”. The MIC concentrations are superimposable, there is no significant difference and furthermore it cannot be said that the deletion of the gene is the only factor determining susceptibility to curcumin.

 Where is the originality of the data if they had already been demonstrated? Ranjan and collaborators (2021 and 2024). I suggest the authors to better highlight the originality of the study.

 More bibliographical references would be appropriate.

Author Response

Dear reviewer,

We thank you very much for your attention towards our manuscript. The quality of our work has been improved by your contribution. Please find the modifications highlighted in the revised version of this manuscript.

We have addressed each point raised by you, as follows:

1 - The description of the results is not clear in the abstract and the full term (PDT), photodynamic therapy, should be inserted.
A.: we have modified the abstract accordingly. 

2 - The authors do not explain why curcumin was chosen. The introduction should expand the literature on the subject, describe in more detail the activity of curcumin as an epigenetic modulator and report other examples of natural substances with similar action.

A.: we chose curcumin because: 1 - it has antifungal activity, incuding against Cryptococcus spp; and 2 - data from cancer cells suggest curcumin to affect HDACs activity, an event that could potentiate the effectiveness of classical antifungal treatments, such as fluconazole and PDT. This information is stated in the introduction and also in the discussion. 

3 - “The results presented in table 1 suggest that the hos2 gene reduces, while the hda1 gene increases, the susceptibility of C. neoformans to curcumin”. The MIC concentrations are superimposable, there is no significant difference and furthermore it cannot be said that the deletion of the gene is the only factor determining susceptibility to curcumin.

A.: we agree with you and have modified the text and the tables including the statistical analysis. We have also expressed the mean, now with a significant number added after the decimal point, and standard error of the mean.

4 - Where is the originality of the data if they had already been demonstrated? Ranjan and collaborators (2021 and 2024). I suggest the authors to better highlight the originality of the study.

A.: we have discussed in the "Introduction" and "Results and discussion" sections the differences that we think are pertinent to be cited in this manuscript. Therefore, we think that going beyond this would be inappropriate. 
But I can clarify this point here: our group as focused on the prospection of HDACs inhibitors for combinatory antifungal treatments. The previous studies by Ranjan, published in 2021 and in 2024, had different goals. The first one reported that classical HDAC inhibitors, such as butyrate, increase the susceptibility of Cryptococcus spp to antifungals such as PDT and fluconazol. The second one focused on prospecting possible HDAC inhibitors from Streptomyces spp. extracts, which have never been studied for their HDAC inhibition potential. The present manuscript explores the potential of curcumin to act as a compound that alters HDAC activity thereby modifying the susceptibility of C. neoformans to PDT. This curcumin's acitivity has never been studied in this fungi, and therefore our study is original in the literature, even considering previous works our group have published. We humbly ask the steemed reviewer to notice that our group has been prospecting HDAC inhibitors, and curcumin is a potential candidate as indicated by studies with mammal cancer cells. 
We have added this discussion to the topic "MIC of AlPc-NE in PDT against C. neoformans strains" in "Results and discussion" section. 

5- More bibliographical references would be appropriate.

A.: we added five more references.

Round 2

Reviewer 2 Report

Comments and Suggestions for Authors

Dear respected author,

I'm so appreciative of your clarification about your remarkable study. I think your explanation is clear to answer all my comments. Thank you very much

Best regards,

Mohamad Gazali, PhD